# Controlled-rearing studies of newborn chicks and deep neural networks

**Donsuk Lee**
Department of Informatics
Indiana University
Bloomington, IN 47408
donslee@iu.edu

**Pranav Gujarathi**
Department of Computer Science
Indiana University
Bloomington, IN 47408
pgujarat@iu.edu

**Justin N. Wood**
Departments of Informatics, Psychology, Neuroscience
Indiana University
Bloomington, IN 47408
woodjn@iu.edu

## Abstract

Convolutional neural networks (CNNs) can now achieve human-level performance on challenging object recognition tasks. CNNs are also the leading quantitative models in terms of predicting neural and behavioral responses in visual recognition tasks. However, there is a widely accepted critique of CNN models: unlike newborn animals, which learn rapidly and efficiently, CNNs are thought to be "data hungry," requiring massive amounts of training data to develop accurate models for object recognition. This critique challenges the promise of using CNNs as models of visual development. Here, we directly examined whether CNNs are more data hungry than newborn animals by performing parallel controlled-rearing experiments on newborn chicks and CNNs. We raised newborn chicks in strictly controlled visual environments, then simulated the training data available in that environment by constructing a virtual animal chamber in a video game engine. We recorded the visual images acquired by an agent moving through the virtual chamber and used those images to train CNNs. When CNNs received similar visual training data as chicks, the CNNs successfully solved the same challenging view-invariant object recognition tasks as the chicks. Thus, the CNNs were not more data hungry than animals: both CNNs and chicks successfully developed robust object models from training data of a single object.

## 1 Introduction

After decades of lagging behind the recognition abilities of even young children, machine-learning systems can now rival human adults on challenging object recognition tasks (Krizhevsky et al., 2012). In addition to powering new technologies, modern machine-learning systems are serving as executable, neurally-mechanistic models in psychology and neuroscience (Kriegeskorte & Douglas, 2018). Cognitive scientists have been particularly interested in a class of computer vision models called deep convolutional neural networks (CNNs), which are directly inspired by neurophysiological observations taken from biological visual systems, including a restricted connectivity pattern that resembles the receptive field organization found in the animal visual cortex (Hubel & Wiesel, 1968; Fukushima, 1980; LeCun et al., 1990). These "end-to-end" models learn to recognize objects from

3rd Workshop on Shared Visual Representations in Human and Machine Intelligence (SVRHM 2021) of the Neural Information Processing Systems (NeurIPS) conference, Virtual.

raw high-dimensional sensory inputs (pixels) and produce behavioral categorizations as outputs. Notably, these models produce internal unit response properties at each level of the network that are similar to actual neurophysiological unit responses at the corresponding levels in animal visual systems (Yamins & DiCarlo, 2016). CNNs can also be used to control the activity state of individual neurons and populations of neurons, indicating that these models capture rich causal properties of how biological visual systems process information (Bashivan et al., 2019). These models are built at scale (i.e., they can take any retinal image as input) and can rival the behavioral performance of mature animals across many challenging visual recognition tasks (Schrimpf et al., 2020).

Despite these strengths, there is a lingering worry about using CNNs as models of the brain. The worry relates to the large amount of data needed to train CNNs, compared to the relatively small amount of training data apparently needed by newborn animals. Specifically, the CNNs that have revolutionized computer vision require massive amounts of training data. These models often have hundreds of layers, tens of millions of parameters, and are trained on millions of images across a thousand different object categories. In contrast, newborn animals require small amounts of training data in order to solve challenging perceptual and motor tasks, with many core abilities emerging within the first few days of life (Held & Hein, 1963; Walk et al., 1957).

One of the most striking differences between newborn animals and CNNs comes in the domain object recognition. In particular, a number of automated controlled-rearing experiments with newborns chicks have demonstrated that chicks rapidly learn to solve challenging object perception tasks, even in the absence of extensive visual experience with objects. Soon after hatching, newborn chicks are capable of object parsing (Wood & Wood, 2021), visual binding (Wood, 2014), view-invariant object recognition (Wood, 2013; Wood & Wood, 2015a), face recognition (Wood & Wood, 2015b), rapid object recognition (Wood & Wood, 2017), action recognition (Goldman & Wood, 2015), and object permanence (Prasad et al., 2019). All of these abilities emerge when chicks are raised in an environment containing a single object, indicating that newborn visual systems can perform "one-shot" learning without extensive training data with objects. From the perspective of CNNs, this is an impressive feat. CNNs typically require thousands to millions of labeled training images to develop object recognition, whereas newborn chicks develop object perception from visual experience with a single object. Consequently, learning in newborn brains appears to be fast and efficient, whereas learning in CNNs appears to be slow and inefficient. If this conclusion were accurate, then it would place significant constraints on the use of CNNs as models of visual development.

Here, we suggest that the learning gap between animals and machines is not as large as it appears. To compare learning across animals and machines, we performed parallel controlled-rearing experiments on newborn chicks and CNNs. First, we raised newborn chicks in strictly controlled virtual environments and measured the chicks' view-invariant object recognition performance. Second, we created a virtual simulation of the controlled-rearing chambers in a video game engine, and then simulated the visual training data available in the chick's environment by recording the first-person images acquired by an agent moving through the virtual animal chamber. Third, we trained CNNs using that simulated training data and tested their object recognition performance with the same recognition tasks used to test the chicks. Accordingly, the chicks and CNNs were trained with the same visual data and tested with the same tasks, allowing for direct comparison of their learning abilities.

## 2 Method

**Animal experiments & stimuli**   In this work, we focused on the view-invariant recognition task from Wood (2013). After hatching, each chick was moved from the incubator to a controlled-rearing chamber in darkness. The controlled-rearing chamber was equipped with two display walls that were used to display the object animations. The chicks' entire visual object experience was limited to the virtual objects projected on the display walls.

The stimulus set from Wood (2013) consisted of two 3D objects, each of which was shown from 12 different viewpoint ranges (Figure A.1). Each animation displayed the object rotating through a 60° viewpoint range about an axis passing through its centroid, completing the full back and forth rotation every 6s. During the first week of life (Training Phase), the chicks' visual experience was limited to a single virtual object seen from a single viewpoint range. During the second week (Test Phase), the chicks were tested on their ability to recognize that object from the 12 viewpoints (11 novel, 1 familiar), using a two-alternative forced choice test. The imprinted object was shown on one

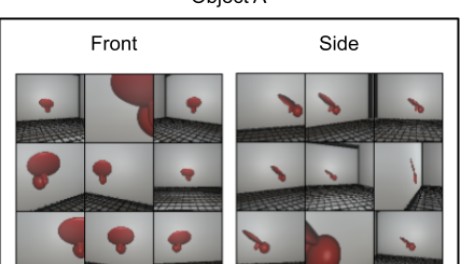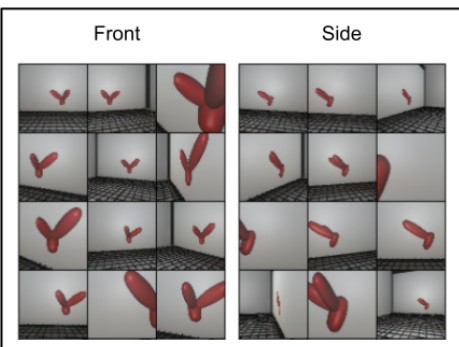

Figure 1: Examples of the simulated training data used to train the CNN models. The dataset was constructed by recording the visual observations of an agent moving within a virtual controlled-rearing chamber.

display wall, and the unfamiliar object was shown on the opposite display wall (see Figure A.2). Test trials were scored as "correct" when the chicks spent a greater proportion of time with their imprinted object and "incorrect" when the chicks spent a greater proportion of time with the unfamiliar object. In order to succeed on the task, chicks needed to learn invariant object representations that generalize across large, novel, and complex changes in the object's appearance.

The chicks performed well on the task, scoring 79% (chance = 50%) when the object was shown from a familiar view and 69% when the object was shown from novel viewpoints (see Wood (2013) for details). Thus, the chicks successfully generated view-invariant representations that generalized across substantial variation in the object's appearance. This study demonstrates that newborn visual systems are capable of building robust object representations from training data of a single object seen from a limited range of views.

**Simulating the training data available to chicks**   In order to mimic the visual experiences of the chicks raised in the controlled-rearing chambers, we constructed an image dataset (Figure 1) by sampling visual observations of an agent moving through a virtual controlled-rearing chamber. The virtual chamber and agent were created with the Unity 3D game engine and the ML-Agents Toolkit (Juliani et al., 2020). The agent received visual input (64×64 pixel resolution images) through a forward-facing camera attached to its head. The agent could move forward or backwards and rotate left or right. We first sampled the visual observations of an agent following a random policy, then programmatically removed images that did not contain an object. The resulting dataset contained 40,000 images (10k images for each of the 4 object animations that the chicks were raised with during the training phase).

**Unsupervised training**   Newborn animals learn through unsupervised (self-supervised) methods. Thus, to directly compare the learning abilities of newborn animals and CNNs, we must use CNNs that learn through unsupervised methods. To this end, we used unsupervised learning algorithms to train the CNNs. As a starting point, we focused on three unsupervised learning algorithms: convolutional autoencoders, simple contrastive learning of representations (SimCLR) (Chen et al., 2020), and "Bring Your Own Latent" (BYOL) (Grill et al., 2020).

To learn useful representations, unsupervised CNNs are trained on self-supervised proxy tasks. Autoencoders learn by first projecting the inputs to lower-dimensional embeddings and then reconstructing the inputs from those embeddings. Contrastive learning methods (SimCLR and BYOL) learn by mapping differently augmented "views" of an image close to each other in the latent embedding space. In SimCLR, positive image pairs are selected by applying two random transforms to an image in the training batch. The rest of the images in the batch are treated as negative examples. Unlike many other contrastive learning methods, BYOL does not rely on negative samples. Instead, it uses two neural networks, referred to as the "online" and "target" networks, that interact and learn from each other. Starting from an augmented view of an image, BYOL trains its online network to predict the target network's representation of another augmented view of the same image.

We used a standard ResNet architecture (He et al., 2016) with 18 layers as the base encoder for all of our CNN models. Importantly, during training, the models only received visual input of a single object rotating through a single viewpoint range (akin to the chicks). For each unsupervised method, we trained models in each of the 4 visual rearing conditions (2 objects presented from 2 viewpoint ranges) presented to the chicks. All models were trained with a batch size of 512 for 500 epochs. Following Chen et al. (2020), we used linear warmup for the first 10 epochs and decayed the learning rate according to cosine annealing schedule without restarts.

**Data augmentation**  Contrastive learning methods rely on data augmentation to generate different "views" of an image. In our experiments, we randomly applied size crops, horizontal flips and color jitters to generate "views" of an image. We used the same data augmentation scheme across all unsupervised methods to provide the same "amount" of training data to each CNN model. Since autoencoders do not require data augmentation for training (unlike SimCLR and BYOL), we trained the autoencoders either with or without data augmentation. Thus, we could test the effect of data augmentation on the quality of representations learned by the autoencoders.

**Baselines**  We also included two baseline models: untrained and supervised CNNs. The untrained CNN was a randomly initialized ResNet-18 encoder. The supervised CNN had the same network architecture as the untrained baseline, but was optimized for a challenging large-scale object classification task (ImageNet; Deng et al., 2009). CNNs trained on the ImageNet dataset achieve high transfer performance on various downstream visual tasks (Huh et al., 2016) and can serve as accurate models of object recognition in mature visual systems (Battleday et al., 2020; Yamins & DiCarlo, 2016). Thus, the transfer performance of an ImageNet trained CNN can serve as a proxy to quantify the performance of an "ideal" observer (mature visual system) on the task.

**Linear evaluation**  With linear classifiers, we evaluated the classification performance of the unsupervised CNNs using the same recognition task that was presented to the chicks. Task performance was assessed by adding a single fully connected linear readout layer on top of the last layer of each trained CNN encoder and then training only the parameters of that readout layer on the binary object classification task. The linear readout layers were optimized for binary cross-entropy loss.

For the linear evaluation, we collected 24,000 simulated visual observations from an agent moving randomly in the virtual chamber (1,000 images for each of 2 objects in 12 viewpoint ranges). The object identities were used as the ground-truth labels.

To evaluate whether the learned representations could generalize across novel viewpoints, we systematically varied the number of viewpoint ranges used to train ($N_{train}$) and test ($N_{test}$) the linear classifiers. We used three different training and test splits, as described below:

- **$N_{train} = 10; N_{test} = 2$** - We first divided the dataset into 6 folds such that each fold contained images of each object rotating through 2 viewpoint ranges. The linear classifiers were cross-validated by training on 5 folds (10 viewpoint ranges) and testing on the held-out fold (2 viewpoint ranges).

- **$N_{train} = 2; N_{test} = 10$** - We divided the dataset into 6 folds in the same way as previously described, but importantly, we inverted the ratio of the training and validation data. Specifically, 1/6 of the images were used for training and 5/6 of the images were used for testing. Thus, the linear classifiers were trained using only 2 viewpoint ranges and tested on the remaining 10 viewpoint ranges.

- **$N_{train} = 1; N_{test} = 11$** - In this extreme case, we split the dataset into 12 folds so that each fold contained images of each object rotating through a single viewpoint range. As such, 1/12 of the images were used for training and 11/12 of the images were used for testing. Thus, the linear classifiers were trained using only 1 viewpoint range from each object and tested on the remaining 11 viewpoint ranges.

For each of the three linear classifier conditions, transfer performance was evaluated by first fitting the parameters of the linear classifier on the training set and then measuring classification accuracy on the held-out test set. We report average cross-validated performance on the held-out images not used to train the linear readout layer.

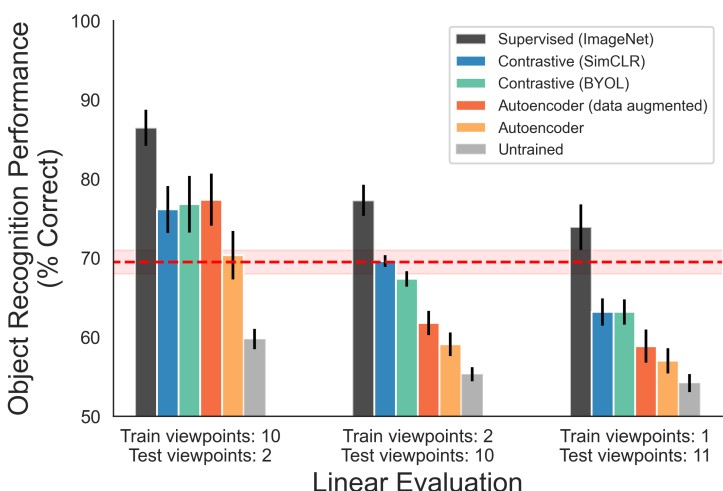

Figure 2: View-invariant object recognition performance of unsupervised and baseline CNNs. The chicks and unsupervised CNNs were only trained with images of a single object seen from a single viewpoint range, and the linear classifiers were cross-validated with different viewpoint ranges in the training versus test sets. Thus, these results reflect the *generalization performance* of the chicks and models across novel views. Like newborn chicks, the unsupervised CNNs could successfully recognize the objects across novel viewpoints, even when the linear classifiers were trained on a small number of viewpoints. The error bars represent standard errors of model performances across validation folds. The red horizontal line shows the chicks' performance on the 2-alternative forced choice task, with the ribbon representing standard error.

## 3 Results and Discussion

Figure 2 shows the view-invariant object recognition performance of the CNNs. We also report the performance of the newborn chicks from Wood (2013). The baseline ImageNet-trained supervised CNN performed the best among all models across all cross-validation schemes, outperforming the chicks by 4.9% points even with the most sparse linear classifiers ($N_{train} = 1$). However, this CNN was trained on a massive dataset (millions of images across 1,000 object categories). Can CNNs still perform well on this task when they receive sparse training data, akin to newborn chicks? All of the unsupervised methods performed on par or better than chicks when the linear classifiers were trained on 10 viewpoint ranges. In more sparse conditions ($N_{train} = 2$ and $N_{train} = 1$), the contrastive learning methods (SimCLR and BYOL) showed comparable performance as the chicks, indicating that CNNs can build invariant object representations from sparse visual input. Across all of the linear classifier conditions, the contrastive learning methods performed on par or better than the autoencoders. This result is consistent with previous studies reporting that contrastive learning methods outperform other classes of unsupervised methods both in downstream classification tasks (Chen et al., 2020) and in tasks that require predicting neural processing in biological visual systems (Zhuang et al., 2021).

To visualize the representational space learned by the CNNs, we used two-dimensional linear discriminant analysis (LDA) (Figure 3). We first extracted CNN features of images randomly sampled from the simulated dataset and then fitted an LDA model using those features as inputs. To explore whether the underlying CNN representations encoded linearly separable information about object identity, we used the combination of 12 viewpoint ranges and 2 object identities (instead of object identities alone) as labels. Although the ground truth of object identity was not explicitly encoded in the labels, the resulting LDA plots for contrastive (SimCLR & BYOL) and supervised features showed clear distinction between the two object categories (green-blue dots vs red-yellow dots). We also show representational dissimilarity matrices (RDMs) for the models in Figure A.3.

**Comparing the training data of newborn chicks and CNNs**   How does the number of images used to train CNNs compare to the number of visual inputs received by newborn chicks? To make

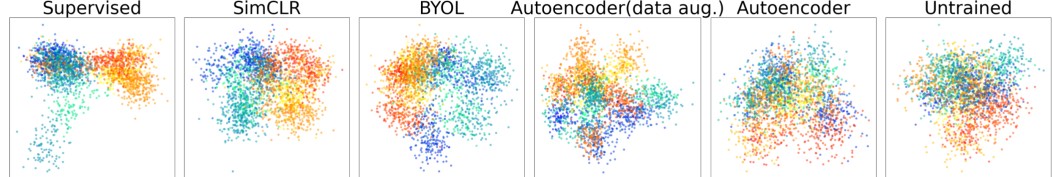

Figure 3: Two-dimensional projections of the feature representations from the untrained, supervised, and unsupervised CNNs. Each point represents a CNN representation of an input image containing a single object. Colors denote the identities and viewpoint ranges of the objects; warm colors (red-yellow) represent Object 1 and cold colors (green-purple) represent Object 2. To create these visualizations, we used linear-discriminant-analysis.

a rough comparison, consider a recent study suggesting that biological visual systems perform a form of predictive error-driven learning every 100 ms, which corresponds to the widely-studied alpha frequency of 10 Hz that originates in deep cortical layers (O'Reilly et al., 2021). If each 100-ms learning window is thought of as a single training image in a computer vision task, then newborn animals acquire 36,000 training images in the first hour after birth. In their first 24 hours of visual experience, newborns will have already acquired nearly a million (864,000) training images. This number is significantly larger than the number of training images (10k) we used to train CNNs in our experiments.

It is important to note, however, that the training data for CNNs in our experiments were augmented with slightly modified copies of already existing images. This increases the effective number of training images received by the CNNs. On the other hand, newborn animals have higher degrees of freedom in their body morphology compared to the artificial agent we used to collect training images. Thus, newborn animals can spontaneously engage in rich data augmentation by acquiring large number of unique object views from diverse bodily configurations. Across animals and machines, heavy data augmentation may be a powerful strategy for learning from raw high-dimensional visual inputs (Bambach et al., 2018).

Finally, in our experiments, the CNNs were trained passively on the images sampled from a randomly-moving agent. In contrast, biological visual systems are embodied, and animals actively interact with their environment to produce their own training data. This active exploration allows newborns to generate their own curriculum and optimize learning (Smith et al., 2018). Future research could close this gap between animals and machines, by embodying CNNs in autonomous artificial agents that collect their own training data from the environment.

**Broader Implications**   Overall, our results show that CNNs can learn to solve view-invariant object recognition tasks from sparse visual input, akin to newborn chicks. For both chicks and CNNs, a visual environment with a single object contains sufficient training data for building a view-invariant object representation. Thus, CNNs might not be as "data hungry" as previously thought. Given that newborn animals and CNNs are capable of similar feats of object recognition after receiving similar sets of training data, we argue that CNNs can serve as models of visual development in animals. Indeed, the initial "proto-architecture" of newborn visual systems shares two core organizational principles with CNNs: both CNNs and newborn visual systems are hierarchically and retinotopically organized (Arcaro & Livingstone, 2021). This finding suggests that the hierarchical and retinotopic architecture of CNNs might be a reasonably good approximation of the "initial state" of newborn visual systems.

There are several advantages to using CNNs as models of visual development because current theories of visual development are incomplete in many ways. First, they are not image computable, requiring a human in the loop to determine the predictions a theory should make in response to a particular stimulus. Second, current theories do not provide a quantitative account of how newborn visual systems learn and respond to raw visual inputs. Third, current theories do not explain why newborn brains develop the way they do. By using CNNs within the framework of goal-driven modeling, we can tackle these problems by building neurally mechanistic, image computable models of newborn visual systems. CNNs can also be tested on a wide range of novel instances of inputs and rigorously falsified in ways that prior models of visual development cannot. Goal-driven modeling with CNNs

has revolutionized the study of mature vision. Likewise, we speculate that this approach can provide a strong computational foundation for the study of visual development.

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

# A  Appendix

## A.1  Animal Experiments & Stimuli

### Object A (Ship)          Object B (Fork)

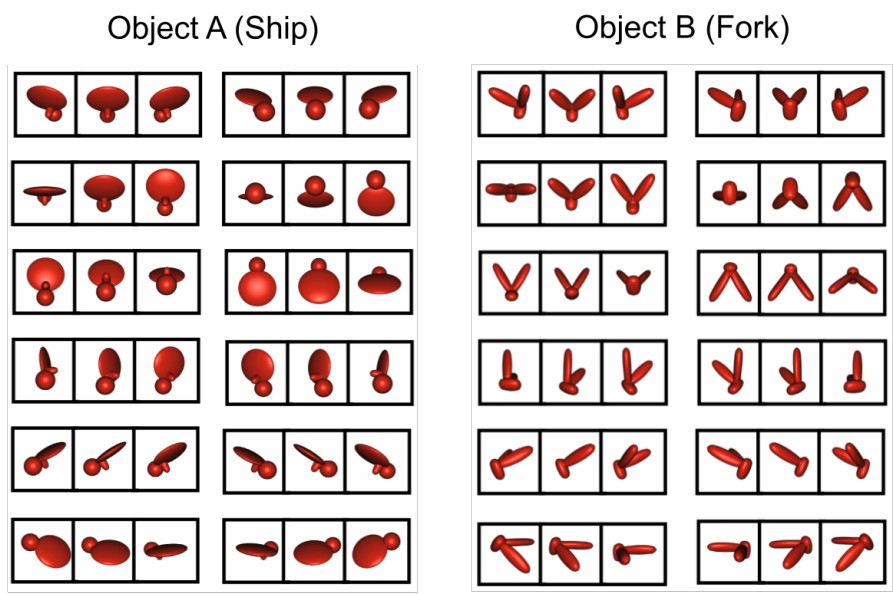

Figure A.1: Object stimuli from Wood (2013). Each triplet shows a virtual object rotating through a 60° viewpoint range. These objects are ideal for studying invariant recognition because changing the viewpoint of an object can produce a greater within-object image difference than changing the identity of the object while maintaining its viewpoint.

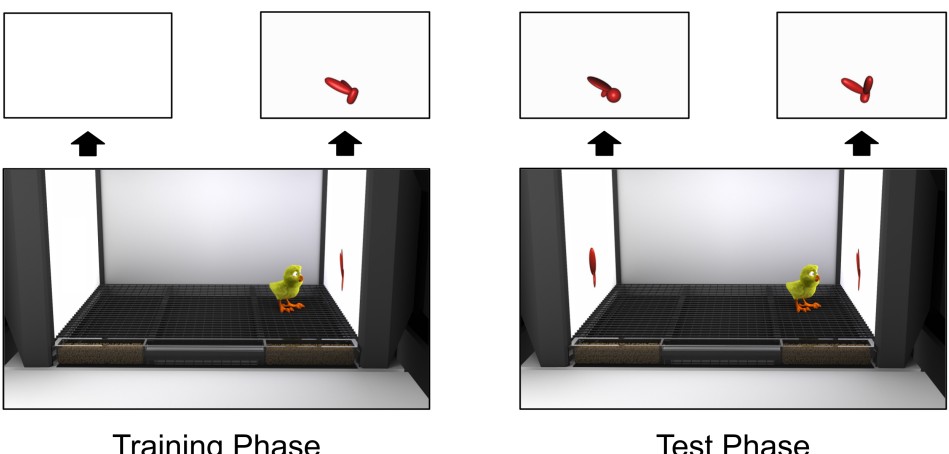

### Training Phase          Test Phase

Figure A.2: Illustration of the controlled-rearing chambers. The chambers contained no objects other than the virtual objects projected on the display walls. During the Training Phase (left), the chicks were exposed to a single virtual object (imprinted object). During the Test Phase (right), the imprinted object was projected on one display wall and an unfamiliar object was projected on the opposite display wall, in a two alternative forced choice test

## A.2   Representation Dissimilarity Matrices

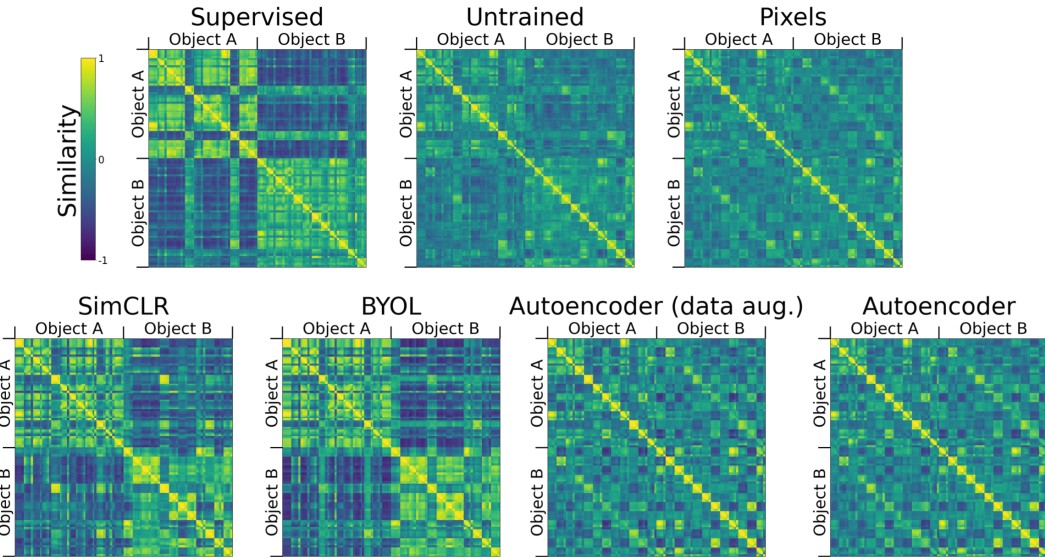

Figure A.3: Representation dissimilarity matrices (RDM) of CNN models. We used pairwise cosine similarity as the distance measure to calculate the RDMs.

