# OpenReview forum: "Controlled-rearing studies of newborn chicks and deep neural networks"
_NeurIPS.cc/2021/Workshop/SVRHM — SVRHM 2021 Oral_

### Official Review · Reviewer_fEL4 · 2021-10-24
**Review of "Controlled-rearing studies of newborn chicks and deep neural networks"**

**Rating:** 7
**Confidence:** 3

**Review:**

This paper compares view-invariant object recognition in newborn chicks and CNNs. The paper argues that, contrary to popular belief, CNNs are not as data hungry as one might think, and perform similar to newborn chicks in object recognition tasks, when training data is matched.

Pros:

•	The idea of comparing deep neural networks to data from controlled-rearing studies is an exciting one, and could be the beginning of a fruitful interdisciplinary field that allows rigorous evaluations on the plausibility of deep neural networks as a model of biological intelligence.

•	The paper includes several useful baseline models.

Cons:

•	models are trained using a dataset of 40,000 images sampled from the virtual rearing chamber. To conclude that the CNN is learning from the same amount of data as newborn chicks, this value needs to be justified. With this in mind, the paper actually shows that CNNs pass the object invariance task with the _variability_ of data available in the virtual chamber, but it does not show that this is done with the same amount of data that newborn chicks have.

•	In addition, the fact that this data is available in the rearing chamber does not imply that newborn chicks are using it to the extent that the CNN might need it.

---

### Official Review · Reviewer_PpG3 · 2021-10-29
**Creative approach that revisit a long-standing question**

**Rating:** 9
**Confidence:** 4

**Review:**

I really appreciated the approach taken by this paper, identifying a productive space for revisiting the long-standing argument surrounding training data in brains and machines. I think the research question is well addressed and the selection of methods was productive.

Comments:
- ResNet-18 is a small DNN but still highly overparametrized model for such a 2AFC task and most random models also perform above chance on it without any prior visual diet (see Fig. 2). I'd find it interesting how a shallower architecture may fare in this scenario (i.e. ResNet10 or even smaller) to further dissect the presented results. For instance, is learning effective despite or due to this overparametrization? More specifically, it would be interesting to introduce a control model that is below chance without training.
- Potentially, the high degree in homogeneity in training data (as was used here) will result in networks that may solve this task but will fail when adapting to new scenarios afterwards, so they may fail to generalize to the "real-world". How does this aspect compare to chicks? E.g. do the chicks that are reared in these controlled environments show less object recognition abilities in richer visual scenarios (e.g. involving textures, occlusions etc.)? One could wonder whether the chick's developmental trajectory and plasticity (which may be not very plastic compared to primates) may be an important difference to the flexibility in DNNs.
- The authors designed a certain sampling of the environment when making the datasets for the models. From the examples (Fig. 1), it seems like this also included variations in altitude. Is that correct and would that be fair to compare to the visual diet of a chick (which presumably do only not yet fly at this age)?
- It would be helpful to also include a measure of variability for the model performance in Fig. 2, for instance, the standard error in performance on the held-out images across folds.
- Omission: "The stimulus set from Wood (2013) consisted [off] two 3D objects"

---

### Official Review · Reviewer_P5tx · 2021-10-29
**A well-written paper comparing chick and DNN performance with similar stimuli**

**Rating:** 7
**Confidence:** 4

**Review:**

This work addresses an interesting problem, of the degree to which DNNs and chicks require a similar quantity of training data. Its strengths are that it uses the same stimuli to train networks and chicks; that it evaluates unsupervised learning algorithms that might be available to chicks; and that clear results are presented.

It is generally well-written, but the motivation for the particular test/train splits is confusing and would benefit from revision. Also, when contrastive learning is introduced (line 105) it would be helpful to mention that it is a SimCLR strategy, and what is used to generate the positive and negative pairs for training. Finally, it is not clear exactly what should be taken from the visualization in Figure 3.

The biggest weakness is that the limitations of the inference are not discussed. Is it actually "degree of learning" that limits performance in the chicks and the DNNs? It is seductive that the unsupervised algorithm gives a similar level of performance to the chicks. But is this really because they have learned the same amount? Presumably, the chick measurements contain other sources of noise (e.g., distraction at test, or motor variability). What is the expected asymptotic learning performance in the chick? And similarly, what limits the DNN. Is it architecture or the quantity of data? It would be helpful to present data at different points in training, to shed light on this. Or, at the very least, a thorough discussion of the limitations of the inference and next stages for investigation would strengthen the work.

---

### Official Review · Reviewer_gEKm · 2021-10-29
**Interesting and comprehensive experiment**

**Rating:** 9
**Confidence:** 4

**Review:**

In this comparative controlled rearing study, the authors approximately matched the visual experience of chicks and CCNs, and compared the view-invariant object recognition capabilities that emerged in the two systems. The work is interesting, the research question is compelling, the experiments are carefully designed (with the logic spelled out), and the paper is thorough and well written. Overall, a clear accept for this venue.

**Specific comments**
* Unsupervised training: When introducing the models, it would be helpful to include their names (e.g. SimCLR). What were the reasons for choosing these two models specifically? Which augmentations were used to train SimCLR?
* Linear evaluation: What type of linear classifier was used (e.g. SVM, multinomial logistic regression with l2 penalty, …)? If it's possible to fit in the text (or otherwise, detail in an appendix, along with additional model training details), it would be useful to provide a brief description (optimization, implementation, etc.)
* Baselines: You might consider moving this section above the "Linear evaluation" section to preempt any concerns about whether an untrained baseline was evaluated, given its importance here.

---

### Decision · Program_Chairs · 2021-11-02

Accept (Oral)